# Isoleucine gate blocks K⁺ conduction in C-type inactivation

Werner Treptow[1,2]*[†], Yichen Liu[3†], Carlos AZ Bassetto[2†‡], Bernardo I Pinto[2], Joao Antonio Alves Nunes[1], Ramon Mendoza Uriarte[2], Christophe J Chipot[2,4,5], Francisco Bezanilla[2,6], Benoit Roux[2]*

[1]Laboratório de Biologia Teórica e Computacional (LBTC), Universidade de Brasília, Brasilia, Brazil; [2]Department of Biochemistry and Molecular Biology, The University of Chicago, Chicago, United States; [3]Department of Neurobiology, The University of Chicago, Chicago, United States; [4]Laboratoire International Associé Centre National de la Recherche Scientifique et University of Illinois at Urbana–Champaign, Unité Mixte de Recherche No. 7019, Université de Lorraine, Université de Lorraine, Vandœuvre-lès-Nancy, France; [5]NIH Center for Macromolecular Modeling and Bioinformatics, Beckman Institute for Advanced Science and Technology, and Department of Physics, University of Illinois at Urbana–Champaign, Urbana, United States; [6]Centro Interdisciplinario de Neurociencia de Valparaíso, Facultad de Ciencias, Universidad de Valparaíso, Valparaíso, Chile

**\*For correspondence:**
treptow@unb.br (WT);
roux@uchicago.edu (BR)

†These authors contributed equally to this work

**Present address:** ‡Department of Physics and Astronomy, The University of Texas at San Antonio, San Antonio, United States

**Competing interest:** The authors declare that no competing interests exist.

**Abstract** Many voltage-gated potassium (Kv) channels display a time-dependent phenomenon called C-type inactivation, whereby prolonged activation by voltage leads to the inhibition of ionic conduction, a process that involves a conformational change at the selectivity filter toward a non-conductive state. Recently, a high-resolution structure of a strongly inactivated triple-mutant channel kv1.2-kv2.1-3m revealed a novel conformation of the selectivity filter that is dilated at its outer end, distinct from the well-characterized conductive state. While the experimental structure was interpreted as the elusive non-conductive state, our molecular dynamics simulations and electrophysiological measurements show that the dilated filter of kv1.2-kv2.1-3m is conductive and, as such, cannot completely account for the inactivation of the channel observed in the structural experiments. The simulation shows that an additional conformational change, implicating isoleucine residues at position 398 along the pore lining segment S6, is required to effectively block ion conduction. The I398 residues from the four subunits act as a state-dependent hydrophobic gate located immediately beneath the selectivity filter. These observations are corroborated by electrophysiological experiments showing that ion permeation can be resumed in the kv1.2-kv2.1-3m channel when I398 is mutated to an asparagine—a mutation that does not abolish C-type inactivation since digitoxin (AgTxII) fails to block the ionic permeation of kv1.2-kv2.1-3m_I398N. As a critical piece of the C-type inactivation machinery, this structural feature is the potential target of a broad class of quaternary ammonium (QA) blockers and negatively charged activators thus opening new research directions toward the development of drugs that specifically modulate gating states of Kv channels.

## Editor's evaluation

This manuscript addresses the molecular mechanism of C-type inactivation observed in a mutant of the Kv2.1-1.2 (Shaker-like) chimeric voltage-gated potassium channel. Previous structural studies using a triple mutant of this channel, which enhance slow inactivation, have demonstrated that inactivation involves a dilation at the outer mouth of the selectivity filter of the channel, leading to a

non-conductive state. Here, based on solid molecular dynamics simulations, corroborated by electrophysiological experiments, the authors conclude that the dilated state on its own is conductive, and that an additional conformational change involving occlusion of the pore by I398 is critical to halt conduction. This important conclusion is thought-provoking and motivates further exploration to evaluate pore dilation and I398 in other Kv channels.

## Introduction

The selectivity filter of voltage-gated potassium (Kv) channels is a specialized molecular structure, responsible for the fast and selective conduction of potassium ($K^+$) over other ionic species. It is well established that the selectivity filter of Kv channels can exist in both conductive and non-conductive states (*Figure 1*). Known as the process of (slow) C-type inactivation (*Hoshi et al., 1990*; *Ostmeyer et al., 2013*), the conductive to non-conductive conformational transition is of great physiological importance as it contributes to fine-tune long-term activity of Kv channels. The canonical conductive conformational state of the filter was first revealed with the crystallographic structure of the prototypical bacterial $K^+$ channel KcsA at high resolution (e.g. PDB id 1BL8 [*Doyle et al., 1998*] or 1K4C [*Zhou et al., 2001*]). Molecular dynamics (MD) studies confirm that ion conduction along this filter conformation is possible and unopposed by large free-energy barriers (*Bernèche and Roux, 2001*; *Kopec et al., 2019*). Since, additional structures have been resolved for other Kv channels of the (eukaryotic) *Shaker* family, broadening our knowledge of the conductive state of the filter (*Long et al., 2007*). Recently, high-resolution structures of Kv channels revealed a novel conformation of the selectivity filter that is partially dilated at its outer end and constricted near its internal face (*Tan et al., 2022*; *Reddi et al., 2022*; *Yangyu et al., 2024*). Because a number of mutations known to strongly enhance the process of C-type inactivation were introduced in the construct used in structural determination (*Perozo et al., 1993*), this 'dilated' conformation has been interpreted as the non-conductive conformational state of the selectivity filter, accounting for the phenomenon of C-type inactivation. Reinforcing that notion, the most recent structure of the wild-type *Shaker* B channel displays the same dilated conformation of the filter at low concentration of external potassium (*Stix et al., 2023*).

The 'dilated' and 'conductive' filter conformations differ markedly. The all-atom root-mean-square deviations (RMSD) of the selectivity filter between the dilated and the conductive conformation is about 3.0 Å, arising mostly from rearrangements of the side chain of the highly conserved tyrosine and aspartic acid along the signature of the selectivity filter TVGYGD (*Figure 1*). Yet, despite the considerable conformational differences, a simple inspection reveals no apparent physical barrier opposing ion conduction along the permeation pathway of the 'dilated' structure, challenging its postulated non-conductivity. The selectivity filter in the dilated conformation is accessible to the intracellular solution via a large open and hydrated vestibular cavity. Incoming ions can bind to sites $S_4$ and $S_3$ and translocate to the extracellular side via a hydrated crevice corresponding to the widened sites $S_2$, $S_1$, and $S_0$. The open and fully hydrated vestibular cavity is clearly conductive in the dilated conformation, which disagrees with previous cysteine modification and blocker-protection assays indicating that the cavity of the channel is changed in the slow-inactivated state compared with the open state (*Panyi and Deutsch, 2007*). Besides, the voltage drop across the selectivity filter through the conductive and dilated conformations is similar (*Figure 1C*). In contrast to classic chemical and peptide blockers or the constricted filter associated with the inactivation of the KcsA bacterial channel that typically inhibit conduction by providing a physical barrier along the permeation pathway (*Ostmeyer et al., 2013*; *Yangyu et al., 2024*; *Eriksson and Roux, 2002*; *Banerjee et al., 2013*; *Cuello et al., 2010*), the dilated conformation does not appear to impose any structural impediment to ionic conduction. These observations beg for a quantitative assessment of the ion conduction properties of the dilated conformation, and the relation of the latter to the functional C-type inactivated state identified in electrophysiological experiments.

## Results

To address these questions, detailed all-atom MD simulations were carried out on the basis of the high-resolution X-ray structure of the kv1.2-kv2.1-3m chimera channel containing the triple mutation W362F, S367T, and V377T (*Reddi et al., 2022*). These mutations enhance C-type inactivation in

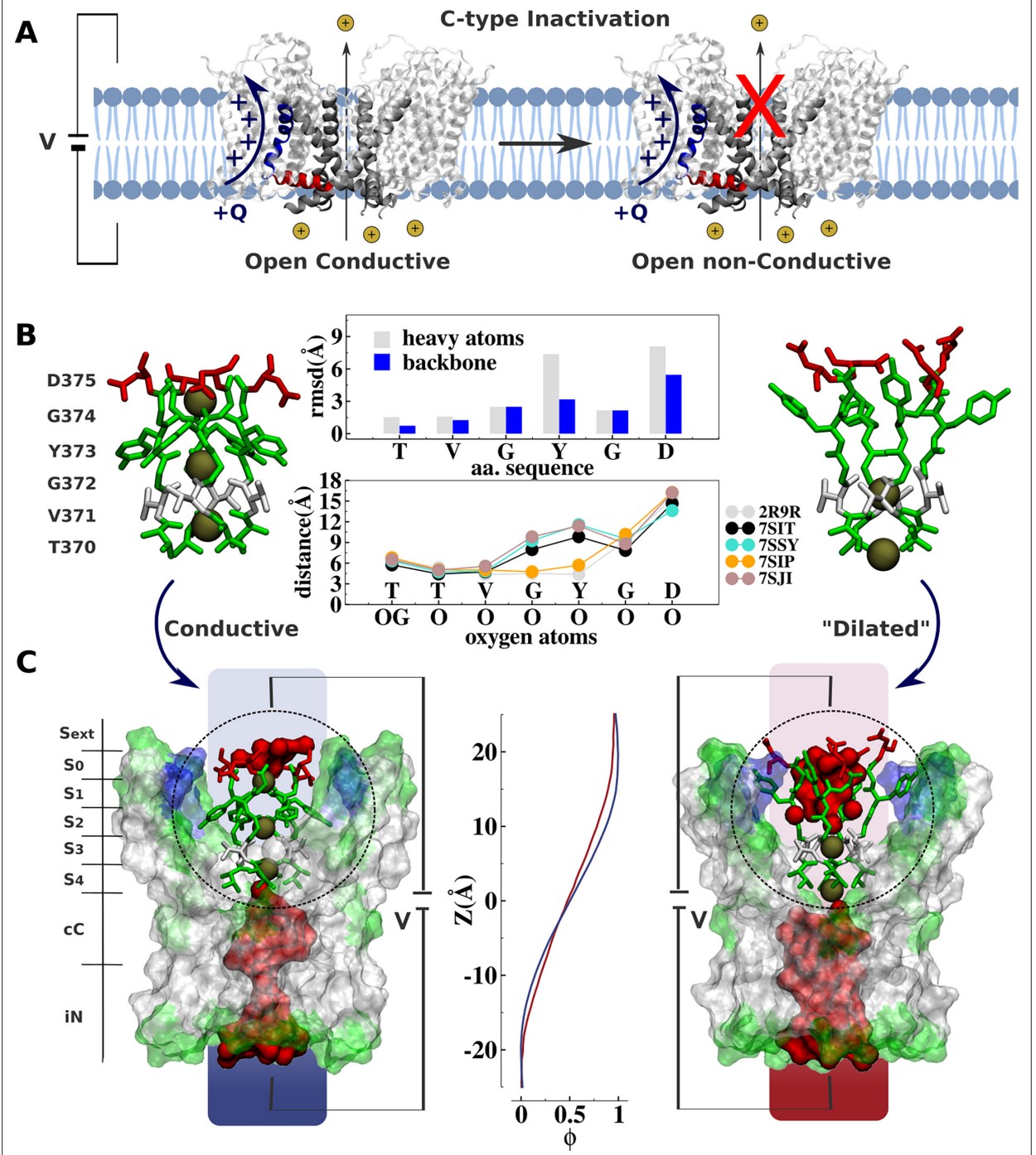

**Figure 1.** Comparative analysis of Kv channel structures. (**A**) Schematic representation of a voltage-gated K+ channel undergoing C-type inactivation, whereby prolonged activation by an external voltage V leads to blockage of ionic conduction across the selectivity filter of the open channel. The voltage-sensor positively charged S4 helix (blue), the S4S5 linker (red), and main-pore S6 helix (silver) are highlighted. (**B**) Structural models of the selectivity filter in the conductive and dilated conformations. The conductive and dilated conformations derive respectively from the high-resolution X-ray structures of the wild-type (*Long et al., 2007*) and triple-mutant (*Reddi et al., 2022*) kv1.2-kv2.1 channel (PDB codes 2R9R and 7SIT). Major structural deviations (root-mean-square deviation [RMSD]) between the selectivity filter conformations are primarily accounted for side-chain rearrangements of Y373 and D375. Despite the side-chain rearrangements of Y373 and D375, the profile of oxygen-oxygen distances between opposing subunits of the selectivity filter indicates that the geometry of sites $S_4$ and $S_3$ in the dilated conformation closely resembles that of the conductive state. For comparison purposes, the profile of oxygen-oxygen distances is also shown along the selectivity filter of the experimental structures of the conductive state of *Shaker* B (*Tan et al., 2022*) (PDB code 7SIP) and the dilated conformations of *Shaker*-W434F (*Tan et al., 2022*) (PDB code 7SJI) and Kv1.3 (*Selvakumar et al., 2022*) (PDB code 7SJ1). (**C**) Electrostatic properties of the conductive and dilated conformations of the selectivity filter. Shown

*Figure 1 continued on next page*

*Figure 1 continued*

are molecular representations of the main-pore S6 segments of the channel, highlighting the permeation pathway along the intracellular entrance (iN), central cavity (cC), and selectivity-filter sites ($S_4$, $S_3$, $S_2$, $S_1$, $S_0$, $S_{ext}$). Hydrated cavities (red) give ionic access to the selectivity filter from the intracellular and extracellular milieu. The dielectric morphology of the protein and waters accounts for a significant voltage drop ($\phi$) across the selectivity filter. The voltage-drop profile along the permeation pathway was computed as described elsewhere (*Souza et al., 2014*), following the charge-imbalance protocol which is a variant of the linear field method (*Roux, 2008*).

the kv1.2-kv2.1 chimera, making the pore similar to the pore of the Shaker channel. To circumvent the uncertainties inherent with MD and achieve robust conclusions, simulations based on two established force fields, AMBER (*Maier et al., 2015*; *Joung and Cheatham, 2008*) and CHARMM36m (*Huang et al., 2017*), were considered. Additional simulations were also carried out using a version of CHARMM36m force field with a few modified interactions to explain the difference in ion conduction with the AMBER force field (Materials and methods, *Supplementary files 1 and 2*) In all the simulations, the channel was embedded in a fully hydrated phospholipid bilayer at 150 mM KCl and simulated with an applied transmembrane (TM) of +200 mV.

The kv1.2-kv2.1-3m channel with the dilated selectivity filter appears remarkably stable during a 10 µs simulation using the AMBER force field (*Figure 2*). The structural RMSD of the selectivity filter relative to the initial X-ray structure (*Reddi et al., 2022*) is less than 2.8 Å, indicative of structural stability (*Figure 2B*, structural deviation R). Simulation of kv1.2-kv2.1-3m with the CHARMM36m force field also illustrates the stability of the dilated conformation over the microsecond timescale, with structural deviation of the selectivity filter less than 2.5 Å (*Figure 2C*, structural deviation R). The average density of the K+ ions along the permeation pathway from the AMBER trajectory is consistent with a predominant occupancy of the sites $S_4$, $S_3$, and $S_{ext}$. Not resolved in the X-ray structure of kv1.2-kv2.1-3m (*Reddi et al., 2022*), the density at the external site $S_{ext}$ results from close interactions of K+ and the carboxylate group of D375. The acidic side chain adopts a relaxed conformation throughout the simulation, in which the carboxylate moiety is fully exposed to the external solution in an orientation similar to that previously reported in the cryogenic electron microscopy (cryo-EM) structure of the homologous Kv1.3 channel with a dilated selectivity filter (*Selvakumar et al., 2022*). According to a hard-knock mechanism at the level of binding sites $S_4$ and $S_3$ (*Figure 2—figure supplement 1*; *Stix*

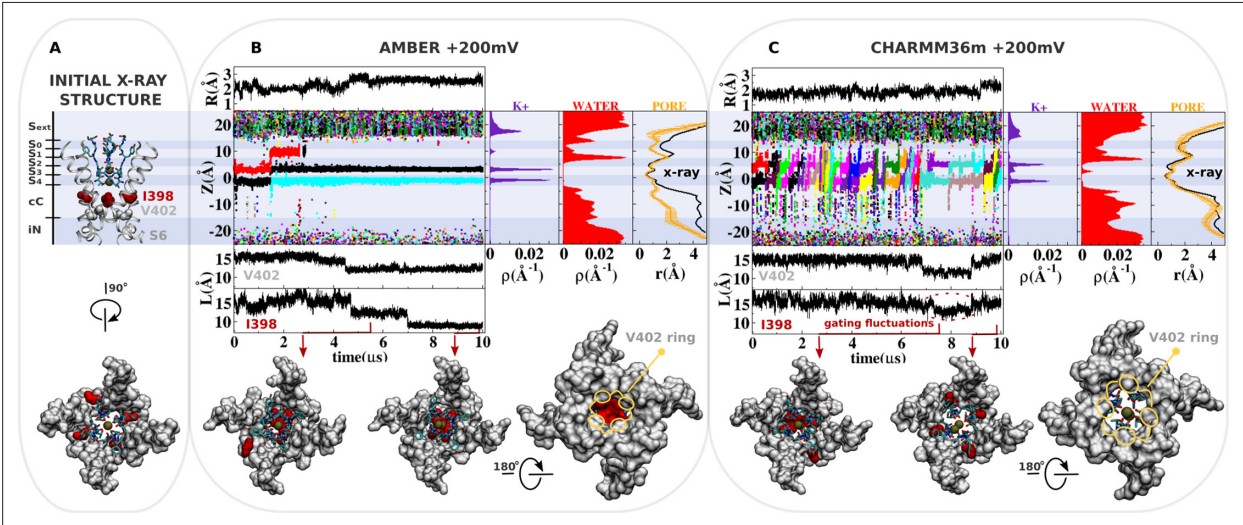

**Figure 2.** Molecular dynamics (MD) simulation of kv1.2-kv2.1-3m at +200 mV. (**A**) Molecular representation of the main pore of the channel, highlighting the initial configuration of the selectivity filter, I398 (red) and V402 (light gray). (**B and C**) Analysis of AMBER and CHARMM36m trajectories. Shown is the structural deviation of the selectivity filter (**R**), the trajectory of K+ ions along the permeation pathway (**Z**), and the intersubunit $C_\beta$-$C_\beta$ separation distance of I398 and V402 (**L**) as a function of simulation time. Inset shows instantaneous configurations of I398 (red arrows). Time averages are the linear density of K+ ions, the linear density of water oxygen, and the pore radius profile (**r**) along the permeation pathway Z.

The online version of this article includes the following figure supplement(s) for figure 2:

**Figure supplement 1.** Ion conduction across the dilated conformation of the selectivity filter of kv1.2-kv2.1-3m.

**Figure supplement 2.** Molecular dynamics (MD) simulation of kv1.2-kv2.1-3m at +200mV.

*et al., 2023*), there is an early voltage-driven K$^+$ conduction event across the selectivity filter within the first ~2 μs of the simulation (*Figure 2B*, trajectory of K$^+$ along the permeation pathway Z). Conduction increases the residence time of the ion in the dilated region of the selectivity filter and accounts for a minor density peak at the level of site S$_1$ (*Figure 2B*, linear density of K$^+$), which is consistent with the resolved electron density of the cation in the experimental structures (*Tan et al., 2022*; *Reddi et al., 2022*; *Yangyu et al., 2024*; *Stix et al., 2023*).

The knock-on mechanism depends on the concentration of incoming ions in the central cavity of the channel. Although the last conduction event takes place around ~2 μs of simulation, note that only after ~4 μs of simulation, the knock-on mechanism is persistently disrupted as ions are excluded from the central cavity of the channel by the isoleucine side chains at position 398. More specifically, ion access to the central cavity of the channel and their subsequent conduction across the selectivity filter completely cease after ~4 μs of simulation time when the side chain of isoleucine at position 398 from all four S6 segments twist toward the central axis of the channel, dehydrating and blocking the permeation pathway immediately beneath the selectivity filter (*Figure 2B*, density of water along the pore axis). The C$_\beta$-C$_\beta$ distance between I398 in opposing subunits decreases over a period of 3–4 μs, reaching a value of about 9 Å—in stark contrast with the initial distance of 15 Å in the X-ray structure (*Reddi et al., 2022*; *Figure 2*, separation distance L of I398). As a consequence, the pore radius becomes locally constricted (≤2 Å), blocking ion conduction (*Figure 2B*, pore radius r) (*Treptow and Tarek, 2006*).

In contrast, ion conduction events are observed in the simulation with the CHARMM36m force field. This observation is in sharp contrast with the conclusion drawn from previous simulations of the dilated conformation of *Shaker* B also based on the CHARMM36m force field, where no ion conduction event was observed even with an applied membrane potential of +300 mV (*Tan et al., 2022*; *Stix et al., 2023*). Importantly, dihedral restraints were applied in these simulations to preclude deviations from the X-ray structure; it is possible that those restraints apparently prevented ion conduction through the open-dilated conformation. Because this is the same conformation of the selectivity filter and the same CHARMM36m force field, suggesting that the key difference in the simulation results is more likely due to different MD simulation conditions used than the very minor differences in the pore domain between Kv1.2-2.1-3m and Shaker. Because the CHARMM36m force field favors the open configuration of the isoleucine gate, the simulation shows that the dilated conformation is highly conductive under membrane depolarization. The distribution of ions along the permeation pathway in the CHARMM36m simulation is distinct from that inferred from the AMBER simulations, as the increased mobility of K$^+$ in the selectivity filter accounts for a more pronounced reallocation of the ionic density from sites S$_4$/S$_3$ to sites S$_2$/S$_1$.

To further clarify the ion conduction properties of the dilated filter and understand the role of isoleucine 398, we simulate the channel with the AMBER force field in the presence of harmonic distance restraint to keep the isoleucine gate in the open configuration (*Figure 2—figure supplement 2*). Compared to the unrestrained simulation with the AMBER force field, a larger number of ions access the central cavity of the channel, triggering two spontaneous conduction events across the selectivity filter in the early ~4 μs of the simulation. The density profile of the K$^+$ ion along the channel axis is distinct from that of the unconstrained simulation. Partial reallocation of the ionic density from sites S$_4$/S$_3$ to sites S$_2$/S$_1$ reflects the increased mobility of K$^+$ in the selectivity filter, thereby corroborating the hypothesis that the dilated conformation of the selectivity filter is conductive when the isoleucine gate is open. This conclusion is further confirmed from additional simulations based on a hybrid CHARMM36m force field referred to as CHARMM36m-NBFIX in which only the ion-carbonyl, ion-water, and water-carbonyl interactions were adjusted to mimic the values from the AMBER force field (*Figure 2—figure supplement 2*, *Supplementary file 2*). While the changes are fairly small and reproduce known behavior for the conductive state of the selectivity filter (Materials and methods, *Supplementary file 3*), the simulations recapitulate the ion conduction properties of the dilated conformation from AMBER, demonstrating how these three key interactions are directly responsible for the observed differences with CHARMM36m. In the CHARMM36m-NBFIX simulation, the local concentration of ions in the central cavity of the channel is significantly smaller than that of the original trajectory. Pronounced density peaks at sites S$_4$/S$_3$ indicates that K$^+$ binds more strongly to the selectivity filter. The combined effect yields a conductive AMBER-like channel, characterized by fewer conduction events per simulation time.

These simulations strongly suggest that there is nothing about the dilated conformation of the filter that inherently impedes ion conduction, and that the conformational motion of I398 is necessary to truly block conduction. Thus, it is the conformational change implicating the isoleucine gate that leads the channel toward a true non-conductive state. As a significant modification of the channel structure, the local rearrangement of I398 seems to be coupled to motions of other regions of the main pore, including V402 at the highly conserved PVP motif (*Figure 2*). Analysis indicates that CHARMM36m favors the open and fully hydrated state of the I398 gate. The average intersubunit $C_\beta$-$C_\beta$ distance of the isoleucine side chains (~16 Å) is close to the reference value in the X-ray structure (~15 Å) most of the simulation time. Gating fluctuations of I398 are, however, clearly observed in the late stages of the simulation and correlate well with the reduction of ions in the central cavity of the channel and with the conduction across the selectivity filter. Particularly important, the CHARMM36m simulation adds support to the assumption that the dilated conformation of the selectivity filter is conductive, and that closure of the isoleucine gate is required to shut down ion transport across the channel.

Despite intrinsic force-field differences with respect to channel conductivity, all three atomistic models support the conclusion that the dilated conformation of the selectivity filter is, by itself, conductive and the isoleucine gate seems to be important to block $K^+$ current across the channel. According to the voltage-driven MD trajectories in which the isoleucine gate is open, the total number of conduction events across the dilated conformation of the selectivity filter over the total simulation time is 44 (ions)/30 μs (*Supplementary file 4*). Based on the number of crossing events per simulation time, the ionic current of a macroscopic population of ion channels in the open-gate dilated state is ~0.2 pA at the TM voltage of +200 mV (single-channel conductance of ~1.17 pS in symmetric 150 mM KCl). In all likelihood underestimation of the channel conductance as a consequence of the well-documented force-field limitations in reproducing the ionic current in Kv channels at low voltages (<300 mV) (*Jensen et al., 2013*), the estimate of ~0.2 pA is still orders of magnitude larger than the measured current in the triple-mutant channel upon C-type inactivation (vide infra), and, therefore, the conductivity properties of the 'dilated' conformation of the selectivity filter cannot explain alone the inactivation of kv1.2-kv2.1-3m under membrane depolarization. Consistent with single-channel measurements (*Yang et al., 1997*), the estimate of ~1.17 pS is actually more comparable to simulation predictions of the single-channel conductance of the conductive selectivity filter i.e., ~3.5 pS in symmetric 300 mM KCl (*Stix et al., 2023*). Beside the dilated conformation of the selectivity filter, the isoleucine gate then appears to be a critical molecular element of the channel machinery, largely implicated in C-type inactivation.

To validate and corroborate the key role of the isoleucine gate in C-type inactivation inferred from the MD simulations, electrophysiology experiments were carried out (Materials and methods, *Figure 3*). In stark contrast with recordings of the kv1.2-kv2.1 chimera channel (*Figure 3A*), C-type inactivation is greatly enhanced in the triple-mutant channel kv1.2-kv2.1-3m, as is evidenced by the fast decay of the ionic current in the ms timescale and the macroscopic current appearing on similar scales as gating current (*Figure 3B*, *Figure 3—figure supplement 1*). However, substitution of the isoleucine by the polar amino acid asparagine, with similar side-chain volume, restores the conductivity of the triple-mutant channel (*Figure 3C*). While not disturbing the current-voltage relationship of the triple-mutant channel, mutation I398N drastically increases ionic conduction without any apparent time-dependent inactivation (*Figure 3D and E*). Importantly, restoration of ion conduction with I398N is not caused by inadvertently stabilizing the selectivity filter in the conductive state, as demonstrated by pore-blocking toxin assays. Agitoxin-II (AgTxII), dendrotoxin, and charybdotoxin (CTX) are potent toxin blockers of Kv channels (*Takacs et al., 2009*), binding to the outer mouth of the channel (*Yangyu et al., 2024*; *Eriksson and Roux, 2002*; *Banerjee et al., 2013*). Such pore-blocking toxins preferentially bind and occlude the conductive conformation of the selectivity filter of Kv channels and are not effective in blocking channels that have W434F-like selectivity filter, as indicated by previous functional studies (*Kitaguchi et al., 2004*) as well as implicit solvent binding free-energy calculations (Materials and methods, *Figure 3F*, *Figure 3—figure supplement 2*, *Supplementary file 5*). Whereas AgTxII blocks the ionic current across the conductive kv1.2-kv2.1 channel by binding to the outer mouth of the filter in the conductive conformation (*Figure 3G*), it fails to bind and occlude the kv1.2-kv2.1-3m channel with its filter mostly in the dilated conformation (*Figure 3H*). Similar effect is observed in Kv1.2-kv2.1-3m_I398N, suggesting the resumed ion permeation is not caused by any stabilization effect of I398N on the selectivity filter and most likely the ions are conducting through the

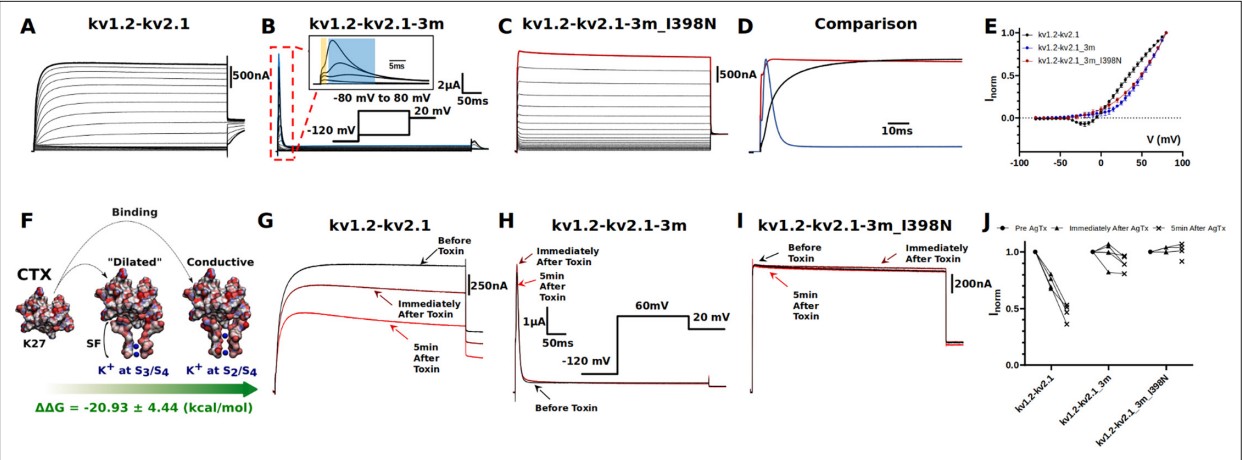

**Figure 3.** Electrophysiology measurements of I398N substitution. (**A, B, C, and D**) Macroscopic current recorded from: (**A**) kv1.2-kv2.1 chimera, (**B**) triple-mutant kv1.2-kv2.1-3m, (**C**) triple-mutant kv1.2-kv2.1-3m with I398N substitution, and (**D**) their respective comparison at +80 mV (the line colors correspond to the colors indicated in **A**, **B**, and **C** of the three mutants at +80 mV). The triple mutation W362F, S367T, and V377T in kv1.2-kv2.1-3m significantly speed up the inactivation process and the gating current could be seen simultaneously with ionic current (shown in inset, with gating current highlighted in yellow and ionic current highlighted in blue). Note the effect of the triple mutation cancel out with the I398N substitution. (**E**) Current-voltage relationship (IV curve). Due to the mixture of gating current and ionic current in kv1.2-kv2.1-3m, only curves shown are taken from the peak current. Voltage-dependent activation relationship for kv1.2-kv2.1, kv1.2-2.1-2m, and kv1.2–2.1-2m_I398N are shown in *Figure 3—figure supplement 3*. (**F**) Net free-energy difference (ΔΔG) involved in the binding of the charybdotoxin (CTX) to the conductive (*Banerjee et al., 2013*) and dilated conformations of the selectivity filter of the kv1.2-kv2.1 chimera channel. ΔΔG indicates a strong preference of CTX to the conductive conformation of the selectivity filter. The same binding preference is found between CTX and kv1.2-kv2.1-3m (*Supplementary file 5*). (**G, H, I, and J**) Effects of agitoxin-II, a more potent CTX analogous (*Takacs et al., 2009*), on kv1.2-kv2.1 chimera, triple-mutant kv1.2-kv2.1-3m, triple-mutant kv1.2-kv2.1-3m with I398N substitution. Clearly, agitoxin binds and blocks the chimera channel while shows minimal influence on triple and triple_I398N, suggesting the selectivity filter in triple_I398N likely also adopts a dilated conformation.

The online version of this article includes the following figure supplement(s) for figure 3:

**Figure supplement 1.** Decay of the ionic current of the triple-mutant kv1.2-kv2.1-3m channel.

**Figure supplement 2.** Estimation of the binding free energy of charybdotoxin (CTX).

**Figure supplement 3.** State dependence of I398N effects.

**Figure supplement 4.** Molecular dynamics (MD) simulation of kv1.2-kv2.1-3m with I398N at +200mV.

**Figure supplement 5.** Primary-sequence conservation throughout the main-pore segments PH, SF, and S6.

dilated filter (*Figure 3I*). Because kv1.2-kv2.1-3m and kv1.2-kv2.1-3m_I398N are not affected by the toxin (*Figure 3J*), the ion conduction in kv1.2-kv2.1-3m_I398N mutant indicates that the permeation pathway must be altered at some location along the permeation pathway other than the selectivity filter. Such structural modifications of the permeation pathway are specific of C-type inactivation since a double mutant (S367T/V377T) of the chimera channel kv1.2-kv2.1-2m that does not inactivate and is blocked by AgTxII, irrespective of the I398N mutation (*Figure 3—figure supplement 3*), suggesting that I398N mutation neither affect the selectivity filter of kv1.2-kv2.1-2m nor kv1.2-kv2.1-3m. The critical C-type inactivation mutation W362F (*Perozo et al., 1993*) is missing in the kv1.2-kv2.1-2m channel, hence its inability to slow-inactivate.

Consistent with experiment, a polar amino acid allows ion conduction by keeping the permeation pathway constitutively hydrated and open. Additional simulations of the triple-mutant channel support this view by revealing that I398N prevents closure of the gate at position 398 (*Supplementary file 1*). *Figure 3—figure supplement 4* shows the AMBER simulation of the triple-mutant channel with the I398N mutation. The conformation of the dilated selectivity filter is stable in presence of the mutation. The average intersubunit $C_\beta$-$C_\beta$ distance of I398N fluctuates between ~11 Å and ~15 Å in the beginning of the AMBER trajectory, before reaching the value of ~9 Å in the final stage of the simulation. Compared to the wild-type simulation, the local pore radius and water density are significantly enhanced under I398N mutation (*Supplementary file 6*). The open and hydrated configuration of the gate in the I398N mutant allows ion conduction across the dilated selectivity filter as long as structural fluctuations of the PVP motif (V402) do not obstruct the permeation pathway, as expected because

that motif is the main gate of the channel. Similar conclusions can be drawn from CHARM36m simulation in which long-lived fluctuations of the mutant gate I398N allows intermittent conduction of ions across the dilated conformation of the selectivity filter over the microsecond timescale.

## Discussion

Taken together, the present computational and experimental results demonstrate that the dilated conformation of the selectivity filter of kv1.2-kv2.1-3m is conductive and that an isoleucine gate is critical to block $K^+$ currents during C-type inactivation of the channel. Judged by the primary sequence conservation of I398 (*Figure 3—figure supplement 5*), the isoleucine gate seems to be relevant for potassium channels that undergo C-type inactivation in general, and, potentially, for other voltage-gated $Na^+$ channels possessing distinct selectivity filters (*Liu et al., 2023*). The action of the isoleucine gate in C-type inactivation is averted by the I398N mutation because the pathway remains hydrated with the polar asparagine side. In the homologous *Shaker* B channel, the single mutation I470C affects the rate of inactivation (*Holmgren et al., 1997*; *Peters et al., 2013*) while the double mutation T449V/I470C (*Olcese et al., 2001*) converts the slow inactivated state into a conductive state. Both mutations alter the slow-inactivation phenotype under long depolarizations, strongly corroborating our findings. Closing of the I398 gate in the dilated conformation of kv1.2-kv2.1-3m involves occlusion of the binding site of internally applied quaternary ammonium (QA) blockers (*Lenaeus et al., 2005*), explaining the previously reported 20-fold decreased affinity of TEA for the inactivated state of *Shaker*-IR compared with that of the open state (*Panyi and Deutsch, 2007*). On the other hand, the mechanism whereby I398N renders the gate constitutively open, not occluding the binding site of QA blockers, is also consistent with the demonstration that I470C in *Shaker*-IR morphs the channel that does not trap QA blockers into one that does (*Holmgren et al., 1997*) and with the fact that MTSEA modifications on the inactivated state are sixfold slower than in the open state of the T449K/I470C *Shaker*-IR (*Panyi and Deutsch, 2007*). Across all these measured effects, the modus operandi of the isoleucine gate is expected to be coupled to the PVP motif, and, therefore, to reflect to some extent the conformational allostery between the selectivity filter and the bundle-crossing region previously reported for C-type inactivation (*Panyi and Deutsch, 2007*; *Cuello et al., 2017*; *Labro et al., 2018*). While these experimental results were indicative of the role of the isoleucine gate its mechanistic significance with regards to the non-conductive C-type inactivation seems to have been overlooked (*Tan et al., 2022*; *Reddi et al., 2022*; *Yangyu et al., 2024*; *Stix et al., 2023*).

It is important to address the approximate and imperfect nature of the atomic models used in the present MD simulations. Even with well-established force fields like AMBER and CHARMM36m, ion conduction through the selectivity filter in the canonical 'conductive' conformation tends to be too small compared to experimental values, especially at physiological voltages (less 50 mV) (*Jensen et al., 2013*). With AMBER, there is observable ion conduction via the hard-knock mechanism (no water molecules between the ions) but a fairly high voltage (200–300 mV) is required to generate a significant ionic current (*Köpfer et al., 2014*). With CHARMM36m, there is also observable conduction via the hard-knock mechanism at high voltages, albeit less so than with AMBER. Thus, the simulated conductance through $K^+$ channels is generally too small, and it is relative to this baseline that conduction through the open dilated conformation of the filter based on the of kv1.2-kv2.1-3m structure must be critically assessed. Here, along the dilated conformation of the selectivity filter, we find that there is no substantial ion conduction with the AMBER force field because the ion binds strongly to the selectivity filter sites $S_4$ and $S_3$. This behavior is reproduced by the CHARMM36m-NBFIX force field, with its AMBER-like ion-carbonyl interactions. However, there is noticeable ion conduction with the CHARMM36m force field because the ion does not bind strongly to the sites $S_4$ and $S_3$. In fact, multiple conduction events across the open-dilated conformation of the triple-mutant chimera channel were observed in the simulation with the CHARMM36m force field, with no applied restraints.

Our physiological experiments demonstrate that by mutating I398 to a polar residue, ion permeation can be resumed in the triple mutation. More importantly, this effect is not a result of reverting C-type inactivation and stabilizing the filter in the conductive state, as shown by the toxin blocking experiments (*Figure 3*, *Figure 3—figure supplement 3*). This implies that the potassium ions are permeating through the dilated filter in kv1.2-kv2.1-3m_I398N, providing strong evidence that I398 is in fact the residue that is responsible for blocking the ion conduction during C-type inactivation instead of the dilated filter. We note that the activation and deactivation become extremely fast in

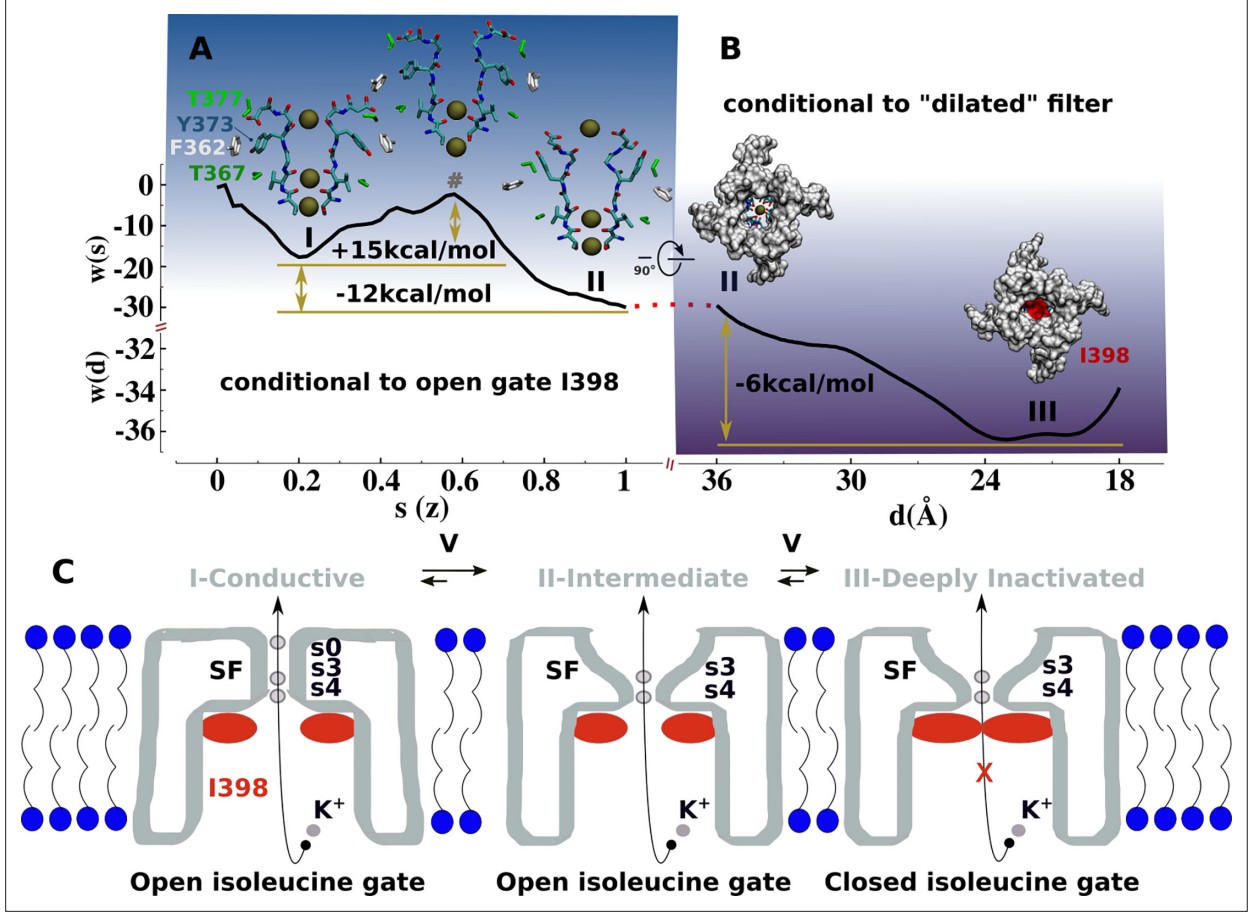

**Figure 4.** Mechanism of C-type inactivation of the triple-mutant channel kv1.2-kv2.1-3m. (**A**) Free-energy profile w(s) along the conformational transition path s connecting the conductive (s=0.2) and dilated (s=1) states of the selectivity filter. The free-energy profile is conditional to the open configuration of the isoleucine gate. (**B**) Free-energy profile w(d) associated to closure of the isoleucine gate. The reaction coordinate d corresponds to the inter-subunit separation distance between β-carbon atoms of I398. The free-energy profile is conditional to the dilated conformation of the selectivity filter. (**C**) C-type inactivation mechanism of the triple-mutant channel inferred from the representative structures of the selectivity filter and isoleucine gate along the free-energy profiles (**A**) and (**B**).

The online version of this article includes the following figure supplement(s) for figure 4:

**Figure supplement 1.** Conformational path between the conductive (**O**) and dilated (**D**) states of the selectivity filter.

**Figure supplement 2.** Convergence analysis of free-energy calculations.

**Figure supplement 3.** Molecular dynamics (MD) simulation of kv1.2-kv2.1-3m at +200 mV and 150 mM NaCl.

the kv1.2-kv2.1-3m_I398N construct which maybe the consequence of rendering the slow-inactivated state conductive, a fact that should be explored further but is beyond the scope of this study. However, the main conclusion still stands: I398 most likely form the barrier for conduction in C-type inactivated channels and the dilated filter itself is not sufficient.

Based on our findings, we propose that the structures of kv1.2-kv2.1-3m (*Reddi et al., 2022*), *Shaker*-W434F (*Tan et al., 2022*), Kv1.2-W366F (*Yangyu et al., 2024*), and *Shaker*-lowK (*Stix et al., 2023*) corresponds to a conductive metastable intermediate on the path toward the true non-conductive C-type inactivated state wherein the permeation pathway is blocked by the isoleucine gate. The free-energy landscape associated with the with C-type inactivation process comprises two sequential transitions between three metastable states: conductive → dilated-intermediate → deeply inactivated. Each transition was characterized by a separate potential of mean force (PMF) calculation (Materials and methods, *Figure 4*, *Figure 4—figure supplements 1 and 2*). The first PMF monitoring the conformation of the selectivity filter while the isoleucine gate is in the open conformation shows two metastable states: conductive and dilated. The transition, which favors the latter state with a downhill free-energy change of −12 kcal/mol, involves a significant conformational change of Y373

from its buried configuration next to T367 in the conductive state toward its externally exposed orientation in the intermediate state resulting from unfavorable close interactions between the polar side chain of Y373 and F362. This is actually a metastable intermediate state that remains conductive. As shown by the second PMF monitoring the closure of the isoleucine gate toward a non-conductive deep inactivated state while the filter remains in dilated conformation shows a downhill change of –6.0 kcal/mol. Notwithstanding the limited accuracies of these computational estimates, the forward and backward free-energy barriers for these two transitions are in qualitative agreement with C-type inactivation of the triple-mutant channel in the sense that kv1.2-kv2.1-3m inactivates substantially faster than recovers from it—at +200 mV, the time constant of C-type inactivation is predicted to be in the same microsecond range of the simulated deep inactivation of the channel (*Figure 3*, *Figure 3—figure supplement 1*). The overall free-energy landscapes pointing toward the greater stability of the 'deep inactivated' state is qualitatively correct.

The new structural insights into the intricate mechanism of C-type inactivation suggests new research directions in the field. Worth of investigation is the experimental observation that *Shaker* B leaks $Na^+$ in the absence or low concentration of $K^+$ in the C-type inactivation state (*Starkus et al., 1997*). AMBER and CHARM36m simulations of kv1.2-kv2.1-3m in presence of 150 mM NaCl show that one or two $Na^+$ ions can stably bind the dilated conformation of the selectivity filter at sites $S_3$, $S_4$, and $S_{ext}$, while favoring the open configuration of the isoleucine gate (Materials and methods, *Figure 4—figure supplement 3* and *Supplementary file 1*). The internal $[Na^+]/[K^+]$ concentration ratio seems to affect the closing of the isoleucine gate with functional implications for sodium leak in C-type inactivation. Also important, it is the more extensive investigation of the structure-function relationship of the inactivation gate according to the amino acid sequence within the pore domain. In particular, note that except for hERG, most studied potassium channels including, Kv1.2, *Shaker*-B, KcsA, and MthK, display either isoleucine, leucine, valine, or phenylalanine at position 398 (*Figure 3—figure supplement 5*). The hERG channel has a tyrosine at position 398 which in the cryo-EM structure (*Wang and MacKinnon, 2017*) is twisted toward the central cavity in the open state. Because tyrosine is bulkier than asparagine, there is a chance that C-type inactivation in hERG also involves constriction of the permeation pathway at position 398 when the selectivity filter is dilated—another fascinating assumption worth of investigation in structural studies. As a piece of the molecular machinery implicated in multiple states, the isoleucine gate might interfere in a state-dependent manner with the mechanism of action of a broad class of QA blockers (*Panyi and Deutsch, 2007*) and negatively charged activators that bind beneath the selectivity filter and operate as master keys to open a variety of $K^+$ channels (*Schewe et al., 2019*). Of particular importance, the understanding of this last aspect could offer a distinct advantage in the development of drugs that modulate gating states of potassium channels more specifically.

# Materials and methods
## Molecular dynamics

The high-resolution X-ray structure of the triple-mutant Kv1.2 channel (kv1.2-kv2.1-3m) was obtained from the Protein Data Bank (PDB code 7SIT) (*Reddi et al., 2022*). The channel structure was embedded in a (POPC) phospholipid bilayer, hydrated by a symmetric 150 mM KCl solution. Using the purpose-built Anton2 supercomputer (*Shaw et al., 2014*), the system was simulated with Desmond at constant temperature 300 K and pressure 1 atm, neutral pH and with applied TM electrostatic potential (*Supplementary file 1*). MD simulations were performed. Equations of motion were integrated using a time step of 2.5 fs and van der Waals interactions were truncated at 12 Å. Ionic currents were driven by application of a constant electric field E across the simulation box to mimic a voltage clamp experiment at the depolarized voltage of +200 mV (corresponding to a TM electric field of 0.043 kcal/mol/Å/e) (*Roux, 2008*). Simulations were performed with three distinct all-atom force fields: AMBER, CHARMM36m, and CHARMM36m-NBFIX. ff14SB version of the AMBER force field (*Maier et al., 2015*) was used in combination with ion parameters by *Joung and Cheatham, 2008*, CHARMM36m (*Huang et al., 2017*) was used with standard ion parameters and CHARMM36m-NBFIX was used with modified ion parameters in which ion-carbonyl interactions were made more attractive to mimic the Joung and Cheatham model. Specifically, all parameters of CHARMM36m are preserved in CHARMM36m-NBFIX except for three critical interactions. In CHARMM36m-NBFIX,

the nonbonded pairwise ion-carbonyl, ion-water, and water-carbonyl Lennard-Jones parameters $E_{min}$ and $R_{min}$ are adjusted (NBFIX) to mimic the value of these three interactions in the AMBER force field. *Supplementary file 2* shows the optimized energy E and distance R for the ion-carbonyl, ion-water, and water-carbonyl interactions in CHARMM36m-NBFIX. Water molecules were described by the TIP3P model (*Jorgensen et al., 1983*). Setup and analysis of the MD trajectories was performed in visual MD (VMD) (*Humphrey et al., 1996*).

## Site-directed mutagenesis and RNA synthesis

Kv1.2-kv2.1 chimera (kindly provided by Eduardo Perozo) was cloned into pMax vector flanked by *Xenopus* β-globin sequence. Mutagenesis was performed utilizing the QuickChange techniques. All the clones were verified with full-length sequencing (Plasmidsaurus). DNA was linearized at the unique PmeI restriction site and then transcribed in vitro using T7 transcription kit (Ambion).

## Channel expression in *Xenopus* oocytes and electrophysiology

Ovaries of *Xenopus laevis* were purchased from XENOPUS1. The follicular membrane was digested by collagenase type II (Worthington Biochemical Corporation) 2 mg/ml supplemented with bovine serum albumin 1 mg/ml. Oocytes were incubated in standard oocytes solution (SOS) containing in mM: 96 NaCl, 2 KCl, 1.8 CaCl$_2$, 1 MgCl$_2$, 0.1 EDTA, 10 HEPES, and pH set to 7.4 with NaOH. SOS was supplemented with 50 mg/ml gentamycin. Stage V-VI oocytes were then selected and microinjected with 50–150 ng of cRNA. Injected oocytes were maintained in SOS solution and kept at 18°C for 1–4 days prior to recordings. Ionic currents were recorded using the cut-open voltage-clamp technique (*Stefani and Bezanilla, 1998*). Voltage-measuring pipettes were pulled using a horizontal puller (P-87 Model, Sutter Instruments, Novato, CA, USA) with resistance between 0.3 and 0.8 MΩ were used to impale the oocytes. Currents were acquired by a setup comprising a Dagan CA-1B amplifier (Dagan, Minneapolis, MN, USA) with a built-in low-pass four-pole Bessel filter for a cutoff frequency of 20 kHz. Using a 16-bit A/D converter (USB-1604, Measurement Computing, Norton, MA, USA) for acquisition and controlled by an in-house software (GPatch64MC), data were sampled at 1 MHz, digitally filtered at Nyquist frequency and decimated for a storage acquisition rate of 100 kHz. Capacitive transient currents were compensated using a dedicated circuit. The voltage clamp was controlled by GPatch64MC and we used the USB-1604 16-bit as the D/A converter. Transient capacitive current was compensated by a dedicated circuit and in some cases, the transients were further minimized by an online P/N protocol holding at –80 mV (*Armstrong and Bezanilla, 1973*). All experiments were performed at room temperature (~17–18°C) in external solution containing: (in mM) 120 potassium methylsulfonate (KMES), 2 calcium hydroxide, 0.1 EDTA, and 10 HEPES, pH = 7.40 (with MES). Internal solution was composed by (in mM): 120 KMES, 10 HEPES, and 2 EGTA, pH = 7.40 (with MES). AgTxII was obtained from Alomone Labs and was titrated to 100 nM in the external solution prior to experiments. The current were elicited prior to the external application of the toxin. The blockage effects were assessed by series of 50 depolarizing pulses (from –120 to +60 mV) every 5 or 10 s. Between experiments, 1% albumin solution (in water) was used to clean the chamber and the bridges. All chemicals used were purchased from Sigma-Aldrich (St. Louis, MO, USA). GraphPad 9 (Prism) and in-house software (Analysis) were used to analyze the data.

## Binding free energy of CTX

The high-resolution X-ray structures of kv1.2-kv2.1 (PDB code 2R9R) (*Long et al., 2007*) and kv1.2-kv2.1-3m (PDB code 7SIT) (*Reddi et al., 2022*) were used as molecular templates for modeling (*Webb and Sali, 2014*) the pore domain of the wild-type, double-mutant (S367T/V377T) and triple-mutant (W362F/S367T/V377T) constructs of the channel in the conductive and 'dilated' conformational states, respectively. Binding of the molecular structure of CTX (*Bontems et al., 1992*) to each of the channel constructs was investigated with HDOCK (*Yan et al., 2020*), according to the condition that Lys27 of CTX is in close proximity to the external entrance of the selectivity filter. The RMSD between docking poses and the X-ray bound configuration of CTX (*Banerjee et al., 2013*) was considered as the structural criterion (RMSD≤5 Å) to select docking solutions best reproducing the bound state of the toxin. At least 10 independent docking solutions were selected for computation of the net free-energy difference (ΔΔG) involved in the binding of CTX to each of the channel constructs and states.

ΔΔG was evaluated according to the continuous implicit solvent calculations of the PB-VDW model used in previous studies (*Eriksson and Roux, 2002*). The Poisson-Boltzmann (PB) solvation energy of the ligand-protein bound complex ($\Delta G_{LP}$) was calculated using the Adaptive Poisson-Boltzmann Solver 1.4.1 (APBS) (*Baker et al., 2001*) through a finite-difference scheme, by considering a 240 Å cubed box and a grid of 1.0×1.0×1.0 Å³. By representing explicitly the protein atoms without any charges, a dummy run was first carried out with APBS to generate dielectric, charge, and accessibility maps for the molecule in solution. Following the molecular surface definition, the internal dielectric constant of the protein was set to 15. The electrolyte solution was represented with a dielectric constant of 80 and salt concentration of 100 mM. These maps were then modified for the inclusion of a low-dielectric (∈=2) lipid surrogate. Input files and maps for APBS were generated with APBSmem (*Callenberg et al., 2010*). The van der Waals component of the bare electrostatic energy of the bound complex ($\Delta E_{LP}$) was computed with the CHARMM36m force field (*Huang et al., 2017*) by using the *namdenergy* plugin linked to VMD (*Humphrey et al., 1996*). The van der Waals component was scaled by an empirical factor ($\lambda$ =0.17), intended to resolve the protein-solvent interaction, absent in the implicit solvent representation (*Nandigrami et al., 2022*). Calculations included the two experimentally resolved bound potassium ions at sites $S_2/S_4$ and $S_3/S_4$ of the conductive and 'dilated' conformation of the selectivity filter, respectively. Solvent accessible surface area and entropic contributions (*Gilson et al., 1997*) associated to the binding energy of CTX were assumed to be similar in both conformations of the channel and as such, they were not included in the calculation of the net binding free-energy difference ΔΔG.

## Primary sequence analysis

Primary sequence logos conservation throughout the main-pore segments PH, SF, and S6 were generated with Weblogo3 (*Crooks et al., 2004*) by taking into consideration an HMMER3.0 (*Finn et al., 2011*) generated multiple sequence alignment of 657 unique UniProt sequences.

## Energetics of C-type inactivation

The energetics of C-type inactivation was investigated by means of two PMFs. The first PMF reports the free-energy profile associated to the conformational transition of the selectivity filter between the conductive and dilated states, conditional to an open isoleucine gate. The second PMF reports the free-energy profile associated to closure of the isoleucine gate under the condition of a dilated conformation of the selectivity filter. Both conditions were imposed in the free-energy calculations via soft harmonic restraints of 0.5 kcal/mol/Å² respectively applied to α-carbon atoms of residue I398 and the selectivity filter. PMFs were determined employing the NAMD (*Phillips et al., 2020*) implementation of the well-tempered metadynamics extended adaptive biasing force algorithm (*Fu et al., 2018*; *Fu et al., 2019*), with the corrected *z*-averaged restraint estimator (*Lesage et al., 2017*). Calculations were respectively carried out with CHARMM36m (*Huang et al., 2017*) and the ff14SB version of the AMBER force field (*Maier et al., 2015*).

The free-energy profile associated to the conformational transition of the selectivity filter between the conductive and dilated states was computed with two path-collective variables (PCVs) (*Branduardi et al., 2007*), formed by 22 internal atomic distances. The first PCV, s, corresponds to the path connecting the two end states of the transformation, i.e., a string of discrete intermediate values inferred from an independent targeted MD simulation, whereas the second, orthogonal one, σ, represents the width of the tube embracing the path. The gradient of the free energy was measured along s, while a soft harmonic potential with a force constant of 5 kcal/mol Å² was applied on ⱳ. No time-dependent bias was applied until a threshold of 50,000 samples was reached.

For the free-energy profile underlying the constriction of the pore domain, the collective variable (CV), $d=d_1+d_2$, was defined as the sum of two Euclidean distances separating the β-carbon atom of residue I398 of subunits KCH1 and KCH3, on the one hand, and of subunits KCH2 and KCH4, on the other hand. The reaction pathway, 18≤d≤36 Å, was discretized in bins 0.1 Å wide, wherein samples of the local force acting along the CV were accrued. To minimize nonequilibrium effects, no time-dependent bias was applied until a threshold of 10,000 samples was reached.

## MD simulation of the MthK channel with CHARMM36m, AMBER, and CHARMM36m-NBFIX

Ion conduction through the canonical 'conductive' conformation of the selectivity filter K$^+$ channels was examined for the well-established CHARMM36m and AMBER force fields, as well as the CHARMM36m-NBFIX modified force field. The simulations were carried out on the basis of the very accurate X-ray crystal structure of the MthK channel at 1.45 Å resolution (PDB code 3LDC) (*Ye et al., 2010*). For each force field, a 2 μs simulation was generated. In the simulated system, the channel structure was embedded in a (POPC) phospholipid bilayer, hydrated by a symmetric 400 mM KCl solution. The system was simulated in the NVT ensemble at constant temperature 320 K and pressure 1 atm, neutral pH, and with applied TM electrostatic potential (300 mV). Restraints were applied to the dihedral angels of the backbone of the selectivity filter with a flat-bottom restraint with a 0.159 kcal/mol/degree$^2$ force constant. The following angels were allowed to vary ±10 degrees from the crystal structure dihedral angels, viz. ±10 degrees from Thr59 ($\phi$: 77, $\psi$: 9), Val60 ($\phi$: –63, $\psi$: –45), Gly61 ($\phi$: 48, $\psi$: 52), Tyr62 ($\phi$: –52, $\psi$: –36), and Gly63 ($\phi$: 84, $\psi$: 8). Beyond these ranges the restraints push the dihedrals back. In addition, to the selectivity restraints, harmonic distance restraints were applied on the level of the C-alpha distances of Pro19 and Phe97 to keep the inner gate open during the simulations, with a force constant of 2.39 kcal/mol/Å$^2$ and applied to both adjacent and opposing subunits, viz., Pro19: 25.5 Å for adjacent and 36 Å for opposing subunits, and Phe97: 26 Å for adjacent and 36.6 Å for opposing subunits. The CHARMM36m and CHARMM36m-NBFIX simulations were performed on OpenMM 7.7 (*Eastman et al., 2017*) for 2 μs with frames recorded every 100 ps. A real-space cutoff of 12 Å was used, and the potential was smoothly truncated at the cutoff with a switching function starting at 10 Å. The long-range electrostatics were treated with PME. The constant temperature was maintained with the Langevin dynamics thermostat with 1/ps friction coefficient. The AMBER simulation followed a similar route as the CHARMM36m simulations with the following changes. The simulation was performed with AMBER20 using the PMEMD GPU accelerated MD simulation engine. A real-space cutoff of 9 Å was used.

Our simulations at 300 mV indicate that the MthK structure is conductive at the level of the selectivity filter sites S$_4$ through S$_1$, with an average conduction rate per voltage of 48.24 pS in symmetric 400 mM KCl— significantly below the recorded single-channel conductance of MthK, i.e., ~170 pS in symmetric 150 mM KCl (*Li et al., 2007*) or ~96 pS in symmetric 200 mM KCl (*Zadek and Nimigean, 2006*). Force-field differences in the simulated conduction rate and mechanism of MthK resume are as follows: (1) conduction is accelerated in the AMBER force field via the hard-knock mechanism; (2) conduction is slower in the CHARMM36 force field via the hard-knock mechanism; and (3) conduction is also accelerated in the AMBER-like CHARMM-NBFIX force field via the hard-knock mechanism. Independently, these findings support that CHARMM36m-NBFIX with only modified ion-carbonyl, ion-water, and water-carbonyl interactions is able to reproduce the known AMBER-like ion conduction behavior, including the accelerated conduction rates and hard-knock mechanism.

## Acknowledgements

Helpful discussions with Eduardo Perozo and Leticia Stock are gratefully acknowledged. We thank Gethiely Gasparini for technical assistance with site-directed mutagenesis and RNA synthesis. Anton 2 computer time was provided by the Pittsburgh Supercomputing Center (PSC) through Grant MCB100018P from the National Institutes of Health. The Anton 2 machine at PSC was generously made available by DE Shaw Research. The work was supported by National Council for Scientific and Technological Development CNPq (WT grant number 302089/2019-5 and 200114/2020-4), by the National Institutes of Health Award R01GM030376 (FB) and R35-GM152124 (BR), National Science Foundation Award QuBBE QLCI (NSF OMA-2121044) (FB). BP is a PEW Latin American Fellow (2019).

## Additional information

### Funding

| Funder | Grant reference number | Author |
|---|---|---|
| National Institute of Health Sciences | R35-GM152124 | Benoit Roux |
| National Institute of Health Sciences | R01GM030376 | Francisco Bezanilla |
| National Science Foundation | OMA-2121044 | Francisco Bezanilla |
| Pew Charitable Trusts | Fellow | Bernardo I Pinto |
| National Council for Scientific and Technological Development | 302089/2019-5 | Werner Treptow |
| National Council for Scientific and Technological Development | 200114/2020-4 | Werner Treptow |

The funders had no role in study design, data collection and interpretation, or the decision to submit the work for publication.

### Author contributions

Werner Treptow, Conceptualization, Formal analysis, Investigation, Visualization, Methodology, Writing – original draft; Yichen Liu, Formal analysis, Methodology, Writing – original draft; Carlos AZ Bassetto, Bernardo I Pinto, Joao Antonio Alves Nunes, Investigation, Methodology, Writing – original draft; Ramon Mendoza Uriarte, performed new multi-microseconds simulations; Christophe J Chipot, Conceptualization, Methodology, Writing – original draft; Francisco Bezanilla, Conceptualization, Formal analysis, Supervision, Methodology, Writing – original draft; Benoit Roux, Conceptualization, Resources, Data curation, Formal analysis, Supervision, Funding acquisition, Validation, Investigation, Methodology, Writing – original draft, Project administration

### Author ORCIDs

Werner Treptow (ID) https://orcid.org/0000-0003-4564-3205
Yichen Liu (ID) https://orcid.org/0000-0003-0774-6932
Carlos AZ Bassetto (ID) http://orcid.org/0000-0002-7012-5699
Bernardo I Pinto (ID) https://orcid.org/0000-0003-0200-1069
Christophe J Chipot (ID) https://orcid.org/0000-0002-9122-1698
Francisco Bezanilla (ID) https://orcid.org/0000-0002-6663-7931
Benoit Roux (ID) https://orcid.org/0000-0002-5254-2712

### Ethics

Electrophysiology using Xenopus oocytes, in compliance with protocol at University of Chicago.

### Decision letter and Author response

Decision letter https://doi.org/10.7554/eLife.97696.sa1
Author response https://doi.org/10.7554/eLife.97696.sa2

## Additional files

### Supplementary files

• Supplementary file 1. Molecular dynamics (MD) simulations of triple-mutant channel kv1.2-kv2.1-3m.

• Supplementary file 2. NBFixes for potassium, carbonyl, and water interactions.

• Supplementary file 3. Number of conduction events along simulations of the open-conductive MthK.

- Supplementary file 4. Number of conduction events along molecular dynamics (MD) simulations in which the isoleucine gate is open.
- Supplementary file 5. Binding free-energy difference of charybdotoxin (CTX).
- Supplementary file 6. Comparative analysis of the average properties of the isoleucine gate with and without mutation (I398N).
- MDAR checklist

## Data availability

All data considered in the study, including molecular configurations and scripts for MD simulations, MD trajectories, docking configurations and electrophysiology data, can be downloaded from the Zenodo repository https://doi.org/10.5281/zenodo.10938041.

The following dataset was generated:

| Author(s) | Year | Dataset title | Dataset URL | Database and Identifier |
|---|---|---|---|---|
| Werner T | 2024 | Isoleucine gate blocks K+ conduction in C-type inactivation | https://doi.org/10.5281/zenodo.10938041 | Zenodo, 10.5281/zenodo.10938041 |

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
