## [Editor Report]

This manuscript addresses the molecular mechanism of C-type inactivation observed in a mutant of the Kv2.1-1.2 (Shaker-like) chimeric voltage-gated potassium channel. Previous structural studies using a triple mutant of this channel, which enhance slow inactivation, have demonstrated that inactivation involves a dilation at the outer mouth of the selectivity filter of the channel, leading to a non-conductive state. Here, based on solid molecular dynamics simulations, corroborated by electrophysiological experiments, the authors conclude that the dilated state on its own is conductive, and that an additional conformational change involving occlusion of the pore by I398 is critical to halt conduction. This important conclusion is thought-provoking and motivates further exploration to evaluate pore dilation and I398 in other Kv channels.

---

## [Decision Letter]

**Decision letter after peer review:**

Thank you for submitting your article "Isoleucine gate blocks K^+^ conduction in C-type inactivation" for consideration by *eLife*. Your article has been reviewed by 3 peer reviewers, and the evaluation has been overseen by a Reviewing Editor and Merritt Maduke as the Senior Editor.

Essential revisions:

During collaborative discussion, three reviewers and the reviewing editor identified the following main concerns that the authors should respond to.

1) It is well established that K-selective channels support potassium permeation through tight coordination by the selectivity filter. It is not clear how a dilated pore might allow selective permeation of this ion. The authors need to clarify possible mechanisms regardless of the MD simulation results.

2) The main conclusion of this manuscript is based on the results of long MD simulations with three force fields. Two of the force fields give inconsistent results and this is fixed by simulations with an ad-hoc correction of the CHARM36 force field. There are no simulations carried out with the corrected CHARMM36m-NBFIX force field that indicate that it continues to reproduce known behavior in WT channels. Authors suggest that the difference in simulation results might be due to the application of harmonic restrains in previous simulations. It should be clarified with simulations if the key differences in results are due to properties of the force fields or the setup of the simulation system.

3) In the AMBER simulation presented in Figure 2, permeation cessation does not seem to be correlated with the movement of the I398 as stated in the text. Please clarify.

4) It is stated that in the AMBER simulations, the distance between β carbons of I398 residues is reduced to 9 angstroms and this leads to a less than 2 angstroms constriction. What are the expected dimensions for the I398N mutant channel? Do simulations of this mutant channel show a WT-like permeation behavior or a dilated selectivity filter.

5) The authors claim that the simulations indicate a similar rate of potassium flow for the kv1.2/2.1-3m channels in the dilated state as for WT Shaker channels. However, it has been known for a while that Shaker W434F channels, which are thought to be permanently c-type inactivated, still allow permeation at the same rate as WT, ~13 pS, just with extremely low open probabilities and very short duration (Yang, Yan and Sigworth, 1997). The simulations presented here, and elsewhere, seem to suggest that the permeability of the slow inactivated state(s) is just significantly reduced. This important discrepancy needs clarification.

6) The 3m channel and the 3m-I398N mutant seem to activate are much more positive voltages than the kv1.2/2.1 channel (Figure 3 E) however in the comparison of time courses (Figure 3d) the I398N mutant activates way faster at the same voltage, this seems to be inconsistent. Also, the voltage pulses employed are too short. Kv1.2 channels slow-inactivate over a time course of seconds. It is possible that the I398N mutant still inactivates over seconds. In fact, Figure 3c shows an indication of slow inactivation as compared with kv1.2/2.1 channels. Given that the I398N mutation activates at more positive voltages, inactivation should be assessed at voltages that saturate the open probability for each channel.

*Reviewer #1:*

In the present manuscript, Treptow, Liu, Bassetto Jr and colleagues propose a novel mechanism for how C-type inactivation diminishes ion conduction in voltage-gated potassium (Kv) channels. C-type inactivation is a time-dependent mechanism that manifests as a decrease in the ionic current within the second timescale until it reaches a steady-state with minimal conductance. Although the mechanism of C-type inactivation was originally studied in the Shaker Kv channel (Hoshi et al. 1991 Neuron), the structural basis for a related mechanism of slow inactivation was first studied in the KcsA channel (Chakrapani et al. 2007 JGP, Cordero-Morales et al. 2006 NSMB, Imai et al. 2010 PNAS, Kim et al. 2016 JGP, Maffeo et al. 2012 Chem.Rev., Piasta et al. 2011 JGP, Tilegenova et al. 2017 PNAS, Varga et al. 2007 Biochim Biophys Acta, Cuello et al. 2010 Nature). A consensus based on the large amount of experimental data collected on KcsA pointed to the selectivity filter (SF) as the responsible part of the channel for C-type inactivation, which was proposed to collapse during inactivation. This collapse of the filter was the accepted working model prior to recent cryo-EM structures of Kv channels in C-type inactivated states showing that the filter dilates during inactivation (Reddi et al. 2022 Sci.Adv, Tan et al. 2022 Sci Adv, Selvakumar et al. 2022 Nat Comm, Wu Y et al. 2024 pre-print, Stix et al. 2023 Sci Adv,). As the mechanism of selective K^+^ permeation at high rates has long been established to result from multiple occupancy of the SF (see below), dilation of the filter to remove the outer K^+^ binding sites would be expected to diminish ion permeation during inactivation, an expectation borne out in Molecular Dynamics simulations (Tan et al. 2022 Sci Adv, Stix et al., 2023Sci Adv). The mechanism proposed by the authors in the present paper, however, challenges our current understanding of the general mechanism for C-type inactivation in Kv1 channels by proposing a new residue, an Ile below the SF as the 'true' gate that impedes ion conduction in C-type inactivated channels. The proposed novel mechanism arises from an MD simulation using the AMBER forcefield and the cryo-EM structure of the Kv1.2-2.1 chimera with 3-point mutations (3m). The 3m mutation (W362F, S367T and V377T) aims to render the channel in a non-conducting C-type inactivated state by speeding its inactivation in a fashion similar to W434F in Shaker (Perozo et al. 1993 Neuron, Yang et al. 1997 JGP) but failing to do so, the Kv1.2/2.1-3m channel is still able to conduct ions; a transient, fast inactivating macroscopic current can be seen upon depolarization. The reason for this is that the chimeric channel contains the sequence of Kv1.2 in the SF, which is known to be particularly resistant to inactivation and requiring more than one mutation to achieve a fast-inactivating phenotype (Suarez-Delgado et al. 2020 JGP, Wu et al. 2022 JGP, Reddi et al. 2022 Sci.Advances).

The fact that the authors chose this particular chimeric channel with the 3 point mutations (Kv1.2/2.1-3m) to study the mechanism of ion permeation during C-type inactivation is an odd choice given that block of ion permeation is incomplete in this mutant (Figure 3B) (Reddi et al. 2022 Sci.Advances). Moreover, the data presented in this manuscript do not support the authors' conclusions.

Strengths:

The new idea presented by the authors is provocative and both MD simulations and electrophysiological techniques are appropriate to explore the mechanism of C-type inactivation.

Weaknesses:

1) The authors seem confused why a dilated filter would be less conductive than one containing 4 ion binding sites. Although details about the mechanism of K^+^ permeation across the SF remain incompletely understood (including whether there is 'hard' or 'soft knock-on' between ions to promote permeation), it would seem to be established that K^+^ channels are exquisitely K^+^ selective because the backbone carbonyls replace waters of hydration and that multiple ions bound within the filter repulse each other to promote rapid throughput (Doyle et al. 1998 and Zhou et al. 2001, Morais-Cabral et al. 2001, Zhou and Mackinnon 2003).

2) The authors proposal that I398 is the gate for C-type inactivation is inconsistent with over multiple structures of KcsA and Kv channels with varying propensities to inactivate because that residue never occludes ion permeation in any of those structures, solved by both X-ray crystallography or cryo-EM. The bar should be high for overturning the weight of evidence that KcsA collapses during inactivation or that Kv channels dilate, and would logically require new structures to support the key conclusions in this study.

3) The MD simulations and the permeation events.

The mechanism proposed by the authors arises from the sole finding of Ile398 twisting during MD simulations produced with the Kv1.2/2.1-3m structure and the AMBER forcefield. The authors propose that the twisting of the Ile creates a gate under the SF that blocks ion permeation during C-type inactivation. This MD result was observed using the unrestrained structure and the AMBER forcefield is the hypothesis generator and the only condition where the authors see this conformational change (Figure 2B). The MD simulations using the CHARMM36 forcefield, on the contrary, show permeation events for half the 10 µs simulation (Figure 2C) and no twisting on the I398. This discrepancy is presented by the authors as a property of the different forcefields used, so they use CHARMM36m-NBFIX to approximate the CHARMM36m force field parameters to AMBER, the permeation events reduce, however, the Ile does not flip in this simulation. These inconsistencies are problematic and not adequately justified by the authors.

In addition, to contextualize these permeation events, it is necessary to see how these simulations, with the exact same conditions and force fields would describe ion permeation for the WT channel in a conducting conformation. The WT MD simulations would likely show many more permeation events without flipping of the I398. This direct comparison will help understand which simulation/force field is more representative of the functional state of the channel and put in context how conductive/nonconductive the Kv1.2/2.1-3m channel is. It seems likely that the present results would be qualitatively consistent with simulations of Shaker performed while constraining the structure and showing that dilation diminishes ion permeation The identity of the I398 as a gate

If the I398 residue is a gate and the residue responsible for diminishing ionic flow in the C-type inactivated state, it is rather curious that it has never been seen before given how many simulations have been run on KcsA and Kv channels. Can the authors provide a rationale for supporting their conclusions in light of what has already been done? How many times did the authors observe the conformational change of this residue relative to the amount of MD runs? If the authors constrain the structure as in previous simulations on Shaker, would I398 no longer adopt a conformation that blocks ion permeation? Also, the results in Figure 2 and FigS2 using the AMBER forcefield, seem to disagree. When the authors repeat the MD simulations using the AMBER forcefield restricting the I398 movement and making it 'permeable' the permeation stops after 2 µs with a long residency of K^+^ ions in the pore. This result alone would seem to challenge the authors hypothesis and clearly suggests that the twist of the I398 is not required to stop ion permeation events because the dilated structure alone seems to be doing that. The CHARMM36m-NBFIX simulations show only 3 permeation events during the whole simulation. How do the authors reconcile these results with their conclusions?

4) Functional consequences of the I398N mutation

It is known that mutations at I398 have a strong functional effect in other channels like Shaker (I470) or KcsA (F103), where previous studies have provided support for a key role of this residue in coupling opening of the inner gate with conformational changes in the SF during inactivation. Those studies are considerably more detailed than the present functional studies and would seem to be inconsistent with I398 functioning as a gate. Val substitutes well for Ile, both Cys and Phe are slower and Leu is faster, and no mutations completely disrupt C-type inactivation (Holmgren et al. 1997, Peter CJ et al. Sci Reports 2013, Cuello et al. 2010 Nature 466 203-8, Cuello et al.2010 Nature 466 272-5). How can the authors reconcile their new ideas with these earlier studies and mechanistic ideas about the role of I398? In addition, introducing an Asn, introduces a polar side chain in a hydrophobic region, a radical change that can affect more than just the C-type inactivation of the channel. In order to understand the effect of I398N it is necessary to study the effect of that mutant in isolation without the 3m mutations since it is a new mutation in the context of Kv1.2/2.1. For instance, does it also impair C-type inactivation in the absence of the 3m mutations? Does it affect the permeation of potassium? Does it shift the G-V curve? Without this information it's not possible to fully understand the results presented in Figure 3. Might introducing a polar residue in this region impact ion binding within the SF? Might the mutation alter inactivation by increasing the affinity of K^+^ for the filter? Finally, the MD simulation results with an Asn at position 398 disagree with the proposed mechanisms and the functional data. The electrophysiological experiments show a channel that conducts ions (Figure 3C) however the MD simulations using the AMBER forcefield do not, and the CHARMM36m only shows permeation events for 3 µs out of 10. How do the authors explain these results?

5) Toxin binding to the outer pore

The results presented with AgTxII seem quite preliminary and it's hard to understand how they support the proposal of I398 functioning as a gate. Only a few traces are shown at one toxin concentration rather than time courses to demonstrate that equilibrium has been achieved. It seems that the 3m mutations are somehow altering toxin binding regardless of whether the I398N mutations rescue ion conduction, but how this supports I398 functioning as a gate is unclear. How do the AgTxII results relate to the model proposed for CTX in Fig3F? Are the authors proposing that the outer pore of the SF changes its architecture when the I398N mutation is introduced?

Recommendations for the authors

1) The statement "These simulations strongly suggest that the filter in the dilated conformation can conduct K^+^ ions, and that the conformational motion of I398 is necessary to truly block conduction." As well as "Despite intrinsic force-field differences with respect to channel conductivity, all three atomistic models support that the dilated conformation of the selectivity filter is, by itself, conductive and the isoleucine gate is required to effectively block K^+^ current across the channel." are not fully supported by the MD results since restricting the conformational motion of I398 also blocks conductions as shown in S2, please review these statements.

2) "Gating fluctuations of I398 are, however, clearly observed in the late stages of the simulation and, correlate well with the reduction of ions in the central cavity of the channel and with the conduction across the selectivity filter. Particularly important, the CHARMM36m simulation adds support to the assumption that the dilated conformation of the selectivity filter is conductive, and that closure of the isoleucine gate is required to shut down ion transport across the channel." Please reference these results in the figure, the CHARMM36m forcefield did not show the Ile changing its conformation in the Figures presented.

3) In observance of the result "the estimate of ~0.2pA is still orders of magnitude larger than the measured current in the triple-mutant channel upon C-type inactivation (vide infra), and, therefore, the conductivity properties of the "dilated" conformation of the selectivity filter cannot explain alone the inactivation of kv1.2-kv2.1-3m under membrane depolarization." There is another possible explanation for this discrepancy related to the configuration of the MD simulations. Taken together the variability of the results restraining vs non-restraining the I398, there is a possibility that the calculations obtained from the MD simulations are not representing the C-type inactivated state of the Kv1.2/2.1-3m triple mutant.

4) "Recently, high-resolution structures of Kv channels revealed a novel conformation of the selectivity filter that is partially dilated at its outer end and constricted near its internal face (8-10)". The internal face of the SF architecture of all the cited structures (S3 and S4) can still solve densities for coordinated K ions in their internal face and as stated in the legend of Figure 1 resembles that on the conductive state, arguing against the constricted conformation stated by the authors please review.

5) This statement in the Abstract could be misleading for the reader, if I understand it correctly, it reads as if the electrophysiology measurements demonstrate that the Kv1.2-2.1-3m mutant is conducting, but then is stated that functional experiments show inactivation, please review: "While the experimental structure was interpreted as the elusive non-conductive state, molecular dynamics simulations and electrophysiology measurements demonstrate that the dilated filter of kv1.2-kv2.1-3m, however, is conductive and, as such, cannot completely account for the inactivation of the channel observed in functional experiments".

*Reviewer #2:*

Based on computational analysis of structures of the conductive WT Shaker B and Kv1.2-2.1 chimera and the pore-dilated Shaker-W434F and triple-mutant Kv1.2-2.1 chimera channels, the authors hypothesize that pore-dilation alone cannot account for the non-conductive tendency of these channels in the C-type inactivated state. The authors then go on to analyze the Kv1.2-2.1 triple mutant (kv1.2-kv2.1-3m) by simulation with AMBER and CHARMM36m force fields, and find that under conditions where the pore-lining residue I398 is allowed to relax, the I398 side chains from all four subunits rapidly twist to occlude K^+^ conduction, whereas K^+^ conduction is maintained under conditions where I398 does not occlude the pore, as in the kv1.2-kv2.1-3m crystal structure.

To validate the role of I398 in controlling conduction in pore-dilated channels, the authors introduce the mutation I398N in kv1.2-kv2.1-3m channels and find that the substitution with the hydrophilic asparagine residue effectively abrogates C-type inactivation behavior. The addition of I398N does not appear to act by preventing the pore-dilated conformation, as Agitoxin-II, which strongly blocks the open-conducting but not pore-dilated channels, does not block the kv1.2-kv2.1-3m-I398N channels.

The manuscript follows a logical series of experiments and thoughtful, rigorous analysis. The mechanism presented is supported by computational and electrophysiological data, and underscores a potential role for conformational changes in pore-lining residues in inactivation that may occur in other K^+^ channels.

Results presented in the manuscript make the strong prediction that an asparagine at position 398 should not occlude the pore in the triple-mutant background, and should stabilize conduction even when the pore is "dilated". It should be possible to show this directly with a simulation, and I think such a demonstration would greatly strengthen the manuscript.

*Reviewer #3:*

This manuscript reports on an investigation by Treptow et al. of C-type inactivation in voltage-gated potassium (Kv) channels, a process whereby prolonged voltage activation leads to a nonconductive state. They examined a triple-mutant Kv1.2-Kv2.1 channel to provide a detailed characterization of the dilated conformation of the selectivity filter. This structure was initially thought to represent the nonconductive state. However, molecular dynamics simulations and electrophysiology showed that this dilated state is actually conductive. The study found that effective inactivation involves an additional conformational change at isoleucine residues (I398) in the pore-lining segment S6, which acts as a hydrophobic gate just below the selectivity filter. This mechanism is critical for C-type inactivation and presents new targets for drug development to modulate Kv channel gating states. This work constitutes a significantly novel contribution to our understanding of the mechanism of conduction of potassium ions. As such, I strongly recommend publication.

I have a few comments that the authors could consider for improving the presentation of their results:

1) The conformational free energy landscape is a crucial piece of the story, which gives quantitative substance to the hypotheses tested throughout the work. However, the description of these results appears surprisingly only in the Discussion section as an afterthought. The authors should make an effort to incorporate these results in the body of the results.

2) Related to the previous point is a general lack of details concerning these calculations. For instance, it is imperative to have an idea of the error associated to the estimated free energies.

3) The differential affinity of the toxin for the two selectivity filter conformations is another crucial piece of the puzzle as it enables to unambiguously interpret the effect of the mutation of isoleucine into asparagine. However, the docking and binding affinity calculations are buried in the supplementary information. The authors should consider giving greater space and emphasis to these results in the main text

4) I am intrigued by the massively different behavior shown by the charmm force-field. What is the reason for this? I wonder if the greater stability of the hydrated configuration of the isoleucine side chain is an artifact due to the water model (tip3p does not reproduce water's surface tension, so wetting/dewetting transitions are not expected to be correctly described).

---

## [Author Response]

Essential revisions:During collaborative discussion, three reviewers and the reviewing editor identified the following main concerns that the authors should respond to.1) It is well established that K-selective channels support potassium permeation through tight coordination by the selectivity filter. It is not clear how a dilated pore might allow selective permeation of this ion. The authors need to clarify possible mechanisms regardless of the MD simulation results.

The net conduction through the “dilated” conformation of the channel is greatly affected by the I398 gate that is located along the TM6 segment below the filter on the intracellular side. When the I398 gate is open, conduction can occur through the dilated filter. It is only when the I398 gate is closed that there is zero conduction. We have not estimated the selectivity of ion conduction through the dilated filter, but it seems likely that the selectivity for K^+^ over Na^+^ would be affected. Supporting that notion, AMBER and CHARMM36m simulations of kv1.2-kv2.1-3m in presence of 150mM NaCl show indeed that one or two Na^+^ ions can stably bind the dilated conformation of the selectivity filter at sites S3, S4 and Sext (Figure 4—figure supplement 3). Furthermore, Jiang and co-workers have shown that the multi-ion occupancy nature of the filter affects selectivity in K^+^ channels. By examining the properties of MthK and NaK mutants, they showed that the channel becomes selective only if there are four consecutive binding sites along the filter [S. Ye, Y. Li and Y. Jiang. Novel insights into K^+^ selectivity from high-resolution structures of an open K^+^ channel pore. Nature structural & molecular biology. 17(8):1019-23 (2010); M.G. Derebe, D.B. Sauer, W. Zeng, A. Alam, N. Shi and Y. Jiang. Tuning the ion selectivity of tetrameric cation channels by changing the number of ion binding sites. Proceedings of the National Academy of Sciences of the United States of America. 108(2):598-602 (2011); D.B. Sauer, W. Zeng, S. Raghunathan and Y. Jiang. Protein interactions central to stabilizing the K^+^ channel selectivity filter in a four-sited configuration for selective K^+^ permeation. Proceedings of the National Academy of Sciences of the United States of America. 108(40):16634-9 (2011)].

2) The main conclusion of this manuscript is based on the results of long MD simulations with three force fields. Two of the force fields give inconsistent results and this is fixed by simulations with an ad-hoc correction of the CHARM36 force field. There are no simulations carried out with the corrected CHARMM36m-NBFIX force field that indicate that it continues to reproduce known behavior in WT channels. Authors suggest that the difference in simulation results might be due to the application of harmonic restrains in previous simulations. It should be clarified with simulations if the key differences in results are due to properties of the force fields or the setup of the simulation system.

First, we would like to emphasize that the experimental validation of the computational observations using site-directed mutagenesis and electrophysiology is a critical part of the present study. Results from long MD simulations with different force fields were used to generate mechanistic hypotheses that were subsequently tested with experiments. The experiments would not have been carried out without the information generated by the simulations.

In summary, the main points about ion conduction from the MD simulations are:

1) Ion conduction through the open-dilated conformation of kv1.2-kv2.1-3m is possible in the standard CHARMM36m force field because the ion does not bind strongly to the sites S4 and S3. Multiple conduction events across the open-dilated conformation of the triple-mutant chimera channel were observed in the simulation with the CHARMM36m force field, with no applied restraints.

2) Ion conduction through the open-dilated conformation of kv1.2-kv2.1-3m is possible but slow in the AMBER force field because the ion binds strongly to the selectivity filter sites S4 and S3;

3) Ion conduction through the open-dilated conformation of kv1.2-kv2.1-3m is also possible but slow in the CHARMM36m-NBFIX force field because it has AMBER-ized ion interactions, and the ion binds also strongly in S4 and S3. These results clarify the observations from points #1 and #2.

The observation of a conductive filter in the dilated conformation with the CHARMM36m force field (#1 above) is in sharp contrast to previously reported simulations of *Shaker* B in which the same force field did not display any ion conduction at +300mV (Tan *et al.* 2022 and Stix *et al.* 2023). This previous simulation was carried out with dihedral restraints to preclude structural distortions of the channel; those restraints apparently rendered the open-dilated conformation non-conductive. Because this is the same structure of *Shaker* B and the same CHARMM36m force field, one can conclude with confidence that the key difference in the results is caused by the restraints applied during the simulation. This aspect was clarified in the revised manuscript.

While it is true that “*Two of the force fields give inconsistent results*” (point #1 and #2 above), this statement gives only a partial account of reality. All force fields are approximate and imperfect. We exploit the simulations to attract our attention to the molecular factors that are of functional relevance with respect to C-type inactivation in K^+^ channels. The ultimate goal is to get a better understanding of this process. Sometimes MD simulations give results that are in quantitative agreement with experiments, and sometimes they don’t. For example, it is well known that ion conduction calculated from MD through the selectivity filter in the canonical “conductive” conformation (as in 1K4C) tends to be too small compared to experimental values, especially at physiological voltages (less 50 mV). This is true for all force fields. With the AMBER force field, there is observable conduction via the hard-knock mechanism but at fairly high voltages (200-300 mV). The reasons for this situation are unknown, likely involving complex polarization effects that are not included in any of the classical force fields used in MD simulations (Jensen *et al.*, 2013). So, simulated conductances are generally too small with current force fields, and the present work is not intended to devise an improved force field to simulate ion conduction through Kv channels. Nonetheless, despite the limitations of current force fields, it is possible to use them to gain insight by comparing the conductance of different conformations of the selectivity filter. For instance, using the AMBER force field or the CHARMM36m-NBFIX force field with AMBER-ized ion interactions (point #3 above), the rate of conduction through the “dilated” filter is not significantly smaller than that of the “conductive” filter. Based on this observation, it is reasonable to propose that the “dilated” conformation should indeed be conductive. This isn’t surprising. After all, there is no obvious physical reason supporting the notion that the dilated conformation should be non-conductive (there is no occlusion or constriction of the pre lumen).

In this context, the purpose and significance of the CHARMM36m-NBFIX force field seems to have been misunderstood. We are sorry for the confusion. It is incorrect to think of this as an effort to “correct” the CHARMM36m force field. The purpose of CHARMM36m-NBFIX is not to fix or improve CHARMM36m. Instead, this force field is only meant to serve as a tool to help reveal how a few key interactions (ion-carbonyl, ion-water, water-carbonyl) are directly responsible for the main observed difference in ion conduction between the AMBER and CHARMM36m force fields. Specifically, all parameters of CHARMM36m are kept unchanged in CHARMM36m-NBFIX except for these 3 key interactions, which are modified to mimic the values from the AMBER force field. The changes are fairly small, at most ~2.0 kcal/mol. In particular, the interaction of K^+^ with the backbone carbonyl group in CHARMM36m-NBFIX is slightly stronger than in CHARMM36m by about -1.8 kcal/mol, to match the AMBER force field. But with these simple modifications, the ion conduction properties of K^+^ through the conductive selectivity filter from CHARMM36m-NBFIX essentially recapitulate that from AMBER. Supplementary file 2 was revised to show the minimized energy *E* and distance *R* for these specific interactions in the CHARMM36m-NBFIX force field, with the fine-tuned Lennard-Jones (LJ) parameters *E*min and *R*min.

Regarding the dynamical fluctuations of the I398 gate, our observations from MD are that the gate can close with the AMBER force field, but less so with the standard CHARMM36m force field. Again, this shows that force fields are approximate and imperfect. We suspect that the reasons for this situation are likely to involve subttle differences in the backbone and side chain torsion potentials. Here, in contrast to ion conduction, we have not been able to pin-point the origin of the difference between the AMBER and CHARMM36m force fields that is directly responsible for these observations by creating a slightly modified CHARMM36m force field that would replicate the AMBER behavior. However, to resolve these inconsistencies and make the most of the imperfect information provided by the simulations, the functional importance of the residue at position 398 was then verified experimentally through site-directed mutagenesis and electrophysiology. The experiments supersede the uncertainty from the MD simulations. What matters at the end is the genuine knowledge remaining when considering the totality of the work.

In summary, our MD simulation study explores the conduction of K^+^ across the dilated conformation of the selectivity filter of the triple-mutant chimera channel kv1.2-kv2.1-3m under two distinct configurations of the isoleucine gate I398: (*i*) conductive (open) or (*ii*) non-conductive (closed). Whenever the isoleucine gate is non-conductive (closed), there is no conduction of ions across the dilated conformation of the selectivity filter. Main-text Figure 2 provides us with most compelling structural evidence for this conclusion. In contrast, we do observe conduction of ions across the dilated conformation of the selectivity filter whenever the isoleucine gate is conductive (open). Ion conduction follows a hard-knock mechanism at the level of the selectivity filter binding sites S4 and S3 (*cf.* Figure 2—figure supplement 1), with an average conduction rate per voltage of ~1.17pS at 150mM KCl (*cf.* Supplementary file 4) that compares well to the simulated conductance of the conductive state ~3.5pS at 300mM KCl (Stix *et al.* 2023). Based on these findings, two consensual conclusions were drawn from the simulations: (1) the dilated conformation of the selectivity filter is, by itself, conductive and (2) the I398 gate is required to effectively block K^+^ current across the triple-mutant channel (*cf.* Figure 2—figure supplement 1B).

Additional information about ion conduction with the CHARMM36m-NBFIX force field

As requested by the reviewers, additional information was provided about ion conduction with the CHARMM36m-NBFIX. We focused on the conductive state of K^+^ channels and compared to CHARMM36m and AMBER. For each force field, an additional simulation of 2μs was performed with the MthK channel. This K^+^ channel is chosen because a very high 1.45Å resolution x-ray structure (PDB code 3LDC) is available. The filter is in the canonical “conductive” conformation as in the 1K4C structure of the KcsA channel. The MthK channel was embedded in a (POPC) phospholipid bilayer, hydrated by a symmetric 400mM KCl solution. The system was simulated in the NVT ensemble at constant temperature 320K and pressure 1atm, neutral pH and with applied transmembrane (TM) electrostatic potential (300mV). Restraints were applied to the dihedral angles of the backbone of the selectivity filter with a flat-bottom restraint with a 0.159 kcal/mol/degree^2 force constant. The following angels were allowed to vary +/- 10 degrees from the crystal structure dihedral angles, viz. +/-10 degrees from Thr59 (ϕ:77, ψ: 9), Val60 (ϕ: -63, ψ: -45), Gly61 (ϕ: 48, ψ: 52), Tyr62 (ϕ: -52, ψ: -36), and Gly63 (ϕ: 84, ψ: 8). In addition, to the selectivity restraints, harmonic distance restraints were applied on the level of the C-α distances of Pro19 and Phe97 to keep the inner gate open during the simulations, with a force constant of 2.39 kcal/mol/Å2 and applied to both adjacent and opposing subunits, viz., Pro19: 25.5Å for adjacent and 36Å for opposing subunits, and Phe97: 26Å for adjacent and 36.6Å for opposing subunits. The CHARMM36m and CHARMM36m-NBFIX simulations were performed on OpenMM 7.7 for 2 microseconds with frames recorded every 100 ps. A real-space cutoff of 12 angstroms was used, and the potential was smoothly truncated at the cutoff with a switching function starting at 10 angstroms. The long-range electrostatics were treated with PME. The constant temperature was maintained with the Langevin dynamics thermostat with 1/ps friction coefficient. The AMBER simulation followed a similar route as the CHARMM36m simulations with the following changes. The simulation was performed on AMBER20 using the PMEMD GPU accelerated MD simulation engine. A real-space cutoff of 9 angstroms was used.

The simulations indicate that the MthK structure is conductive at the level of the selectivity filter sites S4 through S1, with an average conductance of 48.24pS in symmetric 400mM KCl—significantly below the recorded single-channel conductance of MthK *ie.*, ~170pS in symmetric 150mM KCl (Li *et al.*, 2007) or ~96pS in symmetric 200mM KCl (Zadek and Nimigean, 2006). At such high concentration (400mM KCl), one would expect a single-channel conductance on the order of 400-500 pS. Force-field differences in the simulated conduction rate and mechanism resume as following: (1) conduction is accelerated in the AMBER force field via the hard-knock mechanism; (2) conduction is slightly slower in the CHARMM36 force field via the hard-knock mechanism; and, (3) conduction is also accelerated in the AMBER-like CHARMM-NBFIX force field via the hard-knock mechanism. Independently, these findings thus support that the CHARMM36m-NBFIX with AMBER-ized ion interactions reproduces the known AMBER behavior of conductive channels, including the accelerated conduction rates and hard-knock mechanism (Köpfer *et al.*, 2014). Details of the MthK simulations and Supplementary file 3 were added and discussed in the revised manuscript.

3) In the AMBER simulation presented in Figure 2, permeation cessation does not seem to be correlated with the movement of the I398 as stated in the text. Please clarify.

The conduction mechanism depends on the concentration of incoming K^+^ ions in the central cavity of the channel. Although the last conduction event takes place around ~2μs of simulation, note that only after ~4μs of simulation, the knock-on mechanism is persistently disrupted as ions are excluded from the central cavity of the channel by isoleucine I398. There is also a phenomenon of dewetting of the central cavity when the I398 side chain starts to block ion conduction through the pore. The main text was revised accordingly.

4) It is stated that in the AMBER simulations, the distance between β carbons of I398 residues is reduced to 9 angstroms and this leads to a less than 2 angstroms constriction. What are the expected dimensions for the I398N mutant channel? Do simulations of this mutant channel show a WT-like permeation behavior or a dilated selectivity filter.

Figure S6 shows the AMBER simulation of the triple-mutant channel with the I398N mutation. The conformation of the dilated selectivity filter is stable in presence of the mutation. The average intersubunit C_β_-C_β_ distance of I398N fluctuates between ~11Å and ~15Å in the beginning of the AMBER trajectory, before reaching the value of ~9Å in the final stage of the simulation. Compared to the wild-type simulation, the local pore radius and water density are significantly enhanced under I398N mutation—as clearly indicated in the revised Supplementary file 6, showing a comparative analysis of the average properties of the isoleucine gate with and without mutation (I398N). The open and hydrated configuration of the gate in the I398N mutant allows ion conduction across the dilated selectivity filter as long as structural fluctuations of the PVP motif (V402) do not obstruct the permeation pathway. Similar conclusions can be drawn from CHARMM36m simulation in which long-lived fluctuations of the mutant gate I398N allows intermittent conduction of ions across the dilated conformation of the selectivity filter over the microsecond timescale (Figure 3-supplement 4 and Supplementary file 6).

5) The authors claim that the simulations indicate a similar rate of potassium flow for the kv1.2/2.1-3m channels in the dilated state as for WT Shaker channels. However, it has been known for a while that Shaker W434F channels, which are thought to be permanently c-type inactivated, still allow permeation at the same rate as WT, ~13 pS, just with extremely low open probabilities and very short duration (Yang, Yan and Sigworth, 1997). The simulations presented here, and elsewhere, seem to suggest that the permeability of the slow inactivated state(s) is just significantly reduced. This important discrepancy needs clarification.

The fact that the conductance of the dilated or conductive conformations calculated from MD are both smaller than the experimental values should not cause confusion. All force fields are imperfect and tend to underestimate the rate of ion conduction at physiological voltages. The key result from MD is that the dilated filter is able to conduct, at a rate similar to that of the conductive conformation of the filter, and what truly blocks conduction completely is the conformational change involving the I398 gate that occludes the pore. Indeed, the concept of a conductive dilated conformation together with a dynamical I398 gate deduced from the present MD simulations provides a simple and compelling explanation for the observation of permeation in *Shaker* W434F at the same rate as WT with extremely low open probabilities and very short duration that was reported by Sigworth and co-workers (1997).

6) The 3m channel and the 3m-I398N mutant seem to activate are much more positive voltages than the kv1.2/2.1 channel (Figure 3 E) however in the comparison of time courses (Figure 3d) the I398N mutant activates way faster at the same voltage, this seems to be inconsistent. Also, the voltage pulses employed are too short. Kv1.2 channels slow-inactivate over a time course of seconds. It is possible that the I398N mutant still inactivates over seconds. In fact, Figure 3c shows an indication of slow inactivation as compared with kv1.2/2.1 channels. Given that the I398N mutation activates at more positive voltages, inactivation should be assessed at voltages that saturate the open probability for each channel.

It is correct that the 3m channel and the 3m_I398N mutant activate at more positive voltages compared to the kv1.2/2.1 channel. We believe this is most likely due to the triple mutation, especially the W to F mutation (Positions W362F) (see Figure 3—figure supplement 3). The activation kinetics is similar between 2m and 2m_I398N, which we believe is the fair comparison to evaluate the effect of I398N. The 3m channel has 3 mutations compared to WT (W362F, S367T and V377T). A fair comparison to evaluate the effects of I398N is to compare with the double mutant without the W362F mutation. The double mutant (2m) and the double mutant (2m_ I398N) have similar biophysical features as to the WT, G-V, kinetics of activation and inactivation. Moreover, the fact that AgTx is able to bind to and block more efficiently the I398N_2m, but not the I398N_3m demonstrates that I398N does not affect the dilated state of the selectivity filter induced by the W362F mutation. These results point to two main conclusions: (1) the W362F mutation affects the channel displacing the G-V to the right, and (2) the I398N does not affect the channel. Same goes for the slow inactivation comparison. While 3m channel inactivates completely within 10ms of depolarization, the addition of I398N completely changes the channel behavior: after 100ms of depolarization, more than 95% of current is sustained. What we tried to demonstrate here is that a dilated filter, facilitated by the triple mutation, itself doesn’t stop the ion permeation. Only with the additional conformational changes at isoleucine 398 can the channel enter into inactivated state and ceases to conduct. We acknowledge that the kinetics of the 3m_I398N channels are faster than the 3m, 2m, 2m_I398N and the Chimera WT. We do not know exactly the mechanism for this. It could be that I398N combined with the 3m affects the coupling between the activation gate and the selectivity filter. A paragraph discussion is added to address this issue.